# Regulation of Autophagy Is a Novel Tumorigenesis-Related Activity of Multifunctional Translationally Controlled Tumor Protein

**DOI:** 10.3390/cells9010257

**Published:** 2020-01-20

**Authors:** Ji-Sun Lee, Eun-Hwa Jang, Hyun Ae Woo, Kyunglim Lee

**Affiliations:** Graduate School of Pharmaceutical Sciences, College of Pharmacy, Ewha Womans University, Seoul 03760, Korea; leejisun78@hanmail.net (J.-S.L.); ehj_@naver.com (E.-H.J.); hawoo@ewha.ac.kr (H.A.W.)

**Keywords:** TCTP, cancer, autophagy

## Abstract

Translationally controlled tumor protein (TCTP) is highly conserved in eukaryotic organisms and plays multiple roles regulating cellular growth and homeostasis. Because of its anti-apoptotic activity and its role in the regulation of cancer metastasis, TCTP has become a promising target for cancer therapy. Moreover, growing evidence points to its clinical role in cancer prognosis. How TCTP regulates cellular growth in cancer has been widely studied, but how it regulates cellular homeostasis has received relatively little attention. This review discusses how TCTP is related to cancer and its potential as a target in cancer therapeutics, including its novel role in the regulation of autophagy. Regulation of autophagy is essential for cell recycling and scavenging cellular materials to sustain cell survival under the metabolic stress that cancer cells undergo during their aggressive proliferation.

## 1. Role of TCTP in Tumorigenesis

TCTP (also known as histamine releasing factor, HRF; fortilin) is a highly conserved and multifunctional protein that participates in diverse biological and disease processes, including cancer [1,2,3]. Many studies have established the diverse roles of TCTP in tumorigenesis. The best characterized function of TCTP is its anti-apoptotic activity [2,4,5,6]. The anti-apoptotic role of TCTP was first identified in a study that showed the overexpression of TCTP in various cancer cells prevented etoposide-mediated apoptosis [7]. TCTP was shown to exert its anti-apoptotic activity by enhancing the stability of Mcl-1, a member of Bcl-2 family proteins [8]. Further, it was shown that depletion of TCTP in mice led to increased apoptosis during embryogenesis and that lethality supported its role in mediating apoptosis [9]. Crystallography studies revealed that the helical domain of TCTP displays similarity to helices H5-H6 of Bax which have been known to regulate mitochondrial membrane permeability during apoptosis. Based on this finding, a novel mechanism for the antiapoptotic activity of TCTP, which involves its localization to the mitochondria and inhibition of Bax dimerization was proposed [9]. TCTP also potentiates the antiapoptotic function of Bcl-xL through its BH3-like domain [10]. Moreover, genomic integrity is maintained by TCTP via its interaction with ataxia-telangiectasia mutated (ATM) and p53 [11,12]. Low-dose γ-irradiation induced localization of TCTP into the nucleus where it exists complexed with other proteins that participate in DNA damage sensing and repair [11].

TCTP is also implicated in metastasis, tumor invasion, and resistance to anticancer therapy [13,14,15,16]. Knockdown of TCTP in colon adenocarcinoma inhibited proliferation, migration and invasion of tumor cells both in vitro and in vivo [17]. Proteomic analysis revealed that the expression of proteins related to cytoskeleton biosynthesis was changed by TCTP knockdown [13]. In colorectal cancer, extracellular TCTP promoted disease progression and liver metastasis through Cdc42/JNK/MMP9 activation [14]. It was also suggested that TCTP regulates metastasis of colorectal cancer by regulation of the high mobility group 1 (HMGB1) and activation of the NF-KB pathway [17], or by regulating epithelial-mesenchymal transition (EMT) via mammalian target of the rapamycin complex 2 (mTORC2)/Akt/GSK3β/β-catenin pathway [15].

High expression of TCTP in a variety of cancers, including cholangiocarcinoma [18], ovarian cancer [19], and glioma [20], correlates with poor prognosis. Immunohistochemical analysis of normal and colorectal cancer tissues revealed that TCTP expression is significantly higher in the tumor tissues. Elevated TCTP was observed from the adenoma stage, indicating that TCTP is induced early in the disease, and further induction in later stages was not observed [21]. In a cohort of breast cancer patients, TCTP levels increased in later stages of cancer with the poor prognosis [22]. A significant association of TCTP in advanced stages of human hepatocellular carcinoma was observed. This study revealed transcriptional regulation mediated by the chromodomain helicase/ATPase DNA binding protein 1-like gene (CHD1L) [23].

TCTP plays multiple roles in the pathogenesis of cancer, including regulation of resistance to cancer therapy, especially including chemotherapy and radiation therapy [16,21,24]. Jung et al. showed that overexpression of TCTP in HeLa cells inhibited cell death by cytotoxic drugs through the inhibition of mitochondria-mediated apoptosis [16]. During the apoptosome formation induced by chemotherapeutic agents, C-terminally cleaved TCTP associates with Apaf-1 in the apoptosome, thus preventing the caspase activation cascade. In colorectal cancer cells, 5-FU and oxaliplatin treatment induced translational upregulation of TCTP via the mTORC1 pathway and decreased TCTP expression increased the sensitivity to the chemotherapeutic agents [21]. Moreover, breast and lung cancer cells with relatively high expression levels of TCTP were insensitive to radiation-induced cell death and response to radiation was influenced by TCTP levels [24].

Because of its anti-apoptotic properties, its role in the regulation of cancer metastasis, and its relevance to disease prognosis, TCTP has become a promising target for cancer therapy (Table 1). Mechanisms underlying how TCTP regulates cellular growth in cancer have been widely studied, but the question whether autophagy may be involved in how TCTP regulates cellular homeostasis has received relatively little attention. In this review, we discuss TCTP’s role in cancer pathogenesis, especially in relation to its novel role in the regulation of autophagy and its potential as a target in cancer therapeutics.

## 2. Autophagy, a Machinery for Cellular Homeostasis

Autophagy is the process that delivers cellular proteins and organelles to lysosomes for digestion by lysosomal hydrolases, thereby facilitating metabolic homeostasis [25,26,27,28,29]. By eliminating redundant proteins and damaged organelles and by providing building blocks and energy, autophagy serves as a recycling system for cell renewal. During metabolic stress, autophagy prevents the accumulation of toxic cellular components, enabling cells to adapt to environmental changes [29]. Autophagy occurs both in physiological stresses such as nutrient deprivation, ER stress, and hypoxia, as well as in pathological states, including infection, cancer, and neurodegeneration. Therefore, the modulation of autophagy as a therapeutic strategy has recently received increasing attention. But whether autophagy is the cause or result of disease is still an unsettled question. While it is an important catabolic survival mechanism, autophagy has also been classified as type II programmed cell death based on the observation that the cytoplasm of dying cells contains plenty of autophagic vacuolization [26]. However, whether autophagy results in cell death or merely accompanies cell death is still debated [30].

During the autophagic process, cells form double-membrane vesicles called autophagosomes. Lysosome then fuses with the autophagosomes to form autolysosomes. This results in the degradation of the cellular components originally captured by the autophagosomes and in the recycling of their building blocks. This process is divided into five steps: initiation, vesicle nucleation, vesicle elongation, vesicle fusion, and cargo degradation [31]. When cells are in a nutrient-rich environment, the mTORC1 complex inhibits autophagy by preventing the formation of the ULK1 complex (Figure 1) [32,33]. In the initiation step, the ULK1 complex dissociates from mTORC1 and activates the class III PI3K complex. In this complex, Beclin 1 works as a scaffold protein to recruit VPS34, ATG15, UV radiation resistance-associated gene protein (UVRAG), and an activating molecule in BECN1-regulated autophagy protein 1 (AMBRA1). The class III PI3K complex is the main mediator of vesicle nucleation by localizing autophagic proteins into phagophores [33,34].

Once the phagophore, a double-membraned opened structure, has captured its cargo material, the autophagosome undergoes maturation by vesicle elongation process, which is regulated by two ubiquitin-like conjugation systems. The first system involves the incorporation of LC3II into a growing double-membrane, and it is regulated by the protease ATG4B and by the E1-like enzyme ATG7 [35]. The other system requires the formation of the ATG12-ATG5·ATG16L1 complex, which is mediated by ATG7 and by the E2-like enzyme ATG10 [36,37]. In the vesicle fusion step, the autophagosome fuses with the lysosome, facilitated by syntaxin 17 (STX17) [38], and the autophagosome contents are degraded due to the action of the lysosomal enzymes. Autophagy plays an indispensable role in these cellular metabolic events because absence or defective autophagy leads to metabolic, neurodegenerative, and infectious diseases [39]. Defective autophagy has also been reported to cause tumorigenesis, facilitated by several protein partners including TCTP. TCTP has been shown to be involved tumorigenesis, but its specific role in the regulation of autophagy, of which little is known, is discussed in Section 6 below. 

## 3. Role of Autophagy in Tumorigenesis

Autophagy plays a dual role in regulating tumorigenesis: (A) suppressing tumor growth and (B) maintaining the survival of cancer cells. Suppression of tumor growth by inhibiting the proliferation of cancer cells was initially considered as the primary role of autophagy [40]. Exogenous expression of Beclin 1 in human breast cancer carcinoma cells induced autophagy and this was associated with decreased cell proliferation and tumorigenesis in nude mice [41]. Studies in a mutant mouse model revealed BECN1 as a haploinsufficient tumor suppressor gene [42] and provided evidence for negative regulation of tumorigenesis by Beclin-1 [43]. As a possible mechanism for tumor suppression by autophagy, it was proposed that autophagy prevents the accumulation of oxidative stress which would result in genomic instability [27]. During autophagy deficiency, accumulated p62, also known as sequestosome 1 (SQSTM-1), prevents Keap1-mediated ubiquitination of Nrf2, thereby activating an antioxidant defense pathway [44,45]. Another proposed mechanism suggests that autophagy suppresses tumorigenesis through inhibition of inflammation. When autophagy is deficient in cancer cells, necrotic cell death induces an inflammatory response and secretion of pro-inflammatory factors such as the HMGB1 protein, eventually leading to tumor growth [25]. Under chronic inflammation, a proliferation of cells occurs to compensate for the damage from inflammation-inducing cells that are susceptible to genetic mutations and oncogenic transformation [46]. Thus autophagy suppresses inflammation-induced tumorigenesis by mitigating cellular damage [47]. Autophagy also suppresses the proliferation of oncogene-expressing cancer cells by inducing senescence, which functions as a barrier to tumorigenesis [48]. Liu et al. showed that downregulation of ATG5 expression enhanced proliferation and prevented oncogene-induced senescence in melanocytes, thus confirming ATG5 as tumor suppressor gene [49]. Autophagic activity is also upregulated during oncogene-induced senescence and overexpression of ULK3, an autophagy-related gene, resulted in senescence [50]. Thus, autophagy was postulated to suppress tumorigenesis by forcing oncogene-expressing cells to exit the cell cycle.

Cancer cells with high metabolic demand due to increased proliferation depend on autophagy to maintain cancer cell survival [29]. Several studies have reported that autophagy supports cancer cell survival, proliferation, and metastasis-related behaviors. Ectopic expression of H-ras*^v12^* or K-ras*^v12^* in nontumorigenic epithelial cells showed increased basal autophagy, which supports the growth of RAS-driven tumors by maintaining functional mitochondria under the nutrient-limited condition [51]. Moreover, when autophagy is blocked by knockout of ATG5 or ATG7 in these cells, tumor growth in vivo is delayed [51]. Studies with a breast cancer mouse model also demonstrated the tumor-promoting role of autophagy [52]. In this study, conditional knockout of the protein, FIP200, essential for autophagy, inhibited breast cancer initiation, progression, and metastasis. Gene expression profiles of FIP200-knocked out mammary tumors revealed increased expression of immune response genes, suggesting that FIP200 deficiency induces anti-tumor immune surveillance that eventually suppresses tumor progression [52]. Moreover, autophagy inhibition by hydroxychloroquine (HCQ) decreased cellular proliferation and metastasis of dormant breast cancer cells. In this study, knockdown of ATG7 reduced the metastatic burden, while knockdown of BECN1 showed no effects, suggesting that dormant breast cancer cells are autophagic, which is dependent on ATG7 [44]. Additionally, autophagy serves as a survival mechanism for cells under tumor-related conditions, such as hypoxia or metabolic stress. Under hypoxic conditions, hypoxia-inducible factor-1α (HIF-1α) induces mitochondrial hypoxia as an adaptive response to metabolic stress, preventing cell death [53]. With regard to cellular adaptation, autophagy provides cancer cells with a strategy to survive under metabolic stress conditions with limited nutrient and oxygen supply.

Overall, the effect of autophagy on tumorigenesis depends on cancer type, stage, and cancer environment [28,54]. Since genomic instability can initiate cancer [55], autophagy can delay tumorigenesis by maintaining genomic integrity. However, once tumors are formed and begin to proliferate, cancer cells themselves take advantage of autophagy to sustain their rapid proliferation.

## 4. Autophagy, a Potential Target for Cancer Treatment

Probably due to the dual role of autophagy in cancer, both as an inhibitor and as an activator, there is an increasing interest to investigate the modulation of autophagy as a potential avenue for cancer therapy. The application of pharmacological modulators of autophagy has shown promise in some in vitro and in vivo studies, but not in others [56,57,58]. Several studies have reported that treatment with anti-cancer drugs resulted in the induction of autophagy in a range of cancer cell lines [59]. In human mammary carcinoma cells, tamoxifen treatment resulted in autophagic vacuole formation, and 3-methyladenine (3-MA), an inhibitor of autophagosome vacuoles, partially inhibited cell death caused by tamoxifen [60], suggesting that autophagy synergized with apoptosis to cause cell death under the tamoxifen treatment. However, more recent findings suggest that autophagy serves as a possible mechanism of tamoxifen resistance in breast cancer cells [61,62,63]. These studies imply that autophagy induced by anticancer therapy may serve as a cellular protection phenomenon.

Besides, autophagy is also related to radiation-induced cancer cell death. Human breast cancer cells, MCF7, are sensitive to standard doses of radiation. However, the accumulation of acidic vesicular organelles, a characteristic feature of autophagy, was observed in surviving cell population after the irradiation and a potent autophagy inhibitor bafilomycin A enhanced apoptosis-related cell death after the irradiation [64]. These findings suggest that autophagy induced by conventional cancer therapy could be a potential mechanism for the development of therapy resistance by providing cells with an alternative survival pathway to avoid apoptotic cell death. Thus, research on autophagy in the context of cancer therapy and considering autophagy inhibitors in combination with conventional cancer therapy could be a strategy to overcome therapy resistance.

Agents that inhibit autophagy have been suggested as anti-cancer agents, and most class III PI3K inhibitors are categorized to this type. Class III PI3K inhibitors mediate autophagosome formation, and agents such as 3-MA, wortmannin, and LY294002 have been shown to inhibit autophagy [65]. 3-MA, in particular, is distinguished from other PI3K inhibitors in that it exerts a dual role in autophagy. In nutrient-rich conditions, 3-MA promoted autophagy flux while still showing inhibitory effects on starvation-induced autophagy [66]. These differential effects on autophagy arise from the divergent effects of 3-MA on PI3K subunits. While blocking by 3-MA of class I PI3K is persistent, its inhibition of class III PI3K is transient. Under nutrient-rich conditions, prolonged treatment with 3-MA continuously blocks class I PI3K and the Akt-mTOR pathway, which inhibits autophagy while sparing class III PI3K, eventually activating autophagy [66]. Since class I PI3K has been known to be a negative regulator of autophagy [67], selecting PI3K inhibitors as autophagy-regulating anti-cancer agents should take into consideration of how they affect each subtype of PI3K.

Also, lysosomotropic agents such as chloroquine (CQ), HCQ, Lys0569, or monensin have been recommended for curing cancer by preventing acidification of lysosome and eventually inhibiting degradation of molecules in autophagosome [68,69]. Indeed CQ and HCQ are registered anti-malarial drugs and have shown anticancer effects when administered in combination with conventional chemotherapy and radiotherapy [70,71,72]. Furthermore, Bafilomycin A interferes with both lysosome acidification and autophagosome-lysosome fusion and this observation may have the potential to lead to the development of anti-cancer agents that overcome resistance to anticancer therapy [73,74]. A potent autophagy inhibitor spautin-1 [75] also has been reported to enhance the potency of imatinib in chronic myeloid leukemia [76].

On the other hand, inducing excessive autophagy is also suggested for treating cancer, especially in cancer cells that are resistant to apoptotic stimuli. The most commonly studied agent for inducing autophagy is rapamycin, which inhibits mTOR, thus activating autophagy by preventing the inactivating phosphorylation of ULK1. Although rapamycin is an FDA-approved drug for cancers, such as renal cancer, its clinical use has limitations because different cancer cells respond differently to rapamycin [77]. Tyrosine kinase inhibitors and histone deacetylase inhibitors are also known to induce autophagy in cancer cells, but further investigations are needed to settle whether autophagy is induced by cells that are resistant to those inhibitors or as a readout of cell death. It is important to establish whether the induced autophagy results in drug resistance or cell death. Therefore, mechanistic studies about current cancer therapy and autophagy are still worthwhile for enhancing therapeutic efficacy, although autophagy plays a complex role in tumorigenesis.

## 5. Autophagy-Mediated TCTP Degradation

The relationship between TCTP and autophagy has not received much investigative interest, and only a few studies have reported the regulation by and effect on TCTP by autophagy. The expression of TCTP is fine-tuned both by transcriptional and translational regulation by a variety of extracellular signals, but its degradation mechanism is poorly understood. Several studies reported that interaction with several proteins, including Hsp27 and Mcl-1, increase the protein stability of TCTP [78,79] and those studies suggested ubiquitin-proteasome mediated degradation as a TCTP protein regulation mechanism. However, recently, a new type of regulation called chaperone-mediated autophagy (CMA) was suggested as a possible mechanism for TCTP degradation [80]. Unlike in macroautophagy where the substrates are sequestered to the vesicles, which then fuse with lysosomes, in CMA, proteins are directly trapped inside vesicles by the invagination of the lysosomal membrane. Only proteins targeted to the lysosome can be taken up into the lumen of the lysosome [81]. Thus CMA works as a unique proteolytic system that degrades specific cytosolic proteins, which permits quality control of proteins. Thus regulation of CMA could be a promising strategy to combat cancer, as shown by several studies [82,83,84]. In mouse embryonic fibroblasts (MEFs), serum starvation decreased cellular TCTP expression and this was found to be independent of macroautophagy and the ubiquitin-proteasome pathway but was dependent on the 70 kDa heat shock cognate protein (Hsc70) and the lysosome-associated membrane protein type 2A (LAMP-2A) [80]. Hsc70 is a cytosolic chaperone that recognizes the KFERQ motif in the amino acid sequence of the substrate protein, which is mandatory for targeting to the lysosome membrane [85]. LAMP-2A works as a lysosomal receptor for the substrate protein–chaperone complex [86]. Moreover, the KFERQ-like motif in TCTP was generated by acetylation of Lys19, and this increased the binding of TCTP to Hsc70 [80].

## 6. The role of TCTP in the Regulation of Autophagy

Since TCTP influences cell survival and proliferation, it has the potential to influence autophagy, most likely through its association with the mTORC1 signaling pathway. Being at the center of a signaling network, the mTORC1 pathway regulates many processes, which are important for cell growth. This pathway also negatively regulates autophagy by competing with adenosine monophosphate-activated protein kinase (AMPK). During glucose starvation, AMPK drives autophagy by phosphorylation of Ser317 and Ser777 in autophagy initiator, ULK1. When sufficient glucose is provided, mTORC1 phosphorylates Ser757 of ULK1, which prevents the interaction between ULK1 and AMPK, eventually blocking autophagy [87]. We have reported that TCTP is able to interfere with these processes at several points [88], and thereby negatively regulates autophagy (Figure 1). TCTP is translationally induced by growth stimulation through the mTORC1 pathway [89], and therefore regulation of cell growth by TCTP may be related to autophagy. Indeed, decreased TCTP expression promoted formation and maturation of autophagosomes, observed by LC3 puncta formation and by co-localization of LC3 with the lysosomal marker, LAMP1 [88]. Further studies revealed that downregulation of TCTP in HeLa cells resulted in reduced levels of downstream effectors of the mTORC1 pathway, including p-EIF-4EBP1, p-RPS6KB, p-ULK1 (Ser757), but increased AMPK phosphorylation [88]. Interestingly, TCTP knockdown decreased the mTORC1 pathway synergistically with rapamycin, a well-known allosteric inhibitor of mTOR [90], suggesting that TCTP might be a potential target to antagonize rapamycin resistance in cancer therapy. TCTP also regulates autophagy in an AMPK-mTORC1-pathway-independent mode by modulating the formation of the Beclin-1 complex [88]. Beclin-1 is a mammalian orthologue of yeast ATG-6 [91] and performs a central role as overall scaffold for the class III PI3K complex [92]. Additionally, Bcl-2 also blocks autophagy by binding to Beclin-1, disrupting the Beclin-1/PI3K complex [93]. TCTP knockdown reduced bcl-2 expression without affecting Beclin-1 expression and thereby promoted Beclin-1-mediated autophagosome formation [88]. Negatively regulated autophagy by TCTP was further confirmed by a mouse model, in which TCTP is haploinsufficiently expressed [88]. Although more detailed mechanistic studies need to be conducted, we can conclude from the present study that TCTP, because it is involved in regulating cellular autophagy, could be a potential target in cancer therapy.

## 7. Conclusions

TCTP participates in a wide variety of cancer-related phenomena including cancer progression, regulation of apoptosis, tumor reversion, and development of resistance to anti-cancer therapy. This suggests that pharmacological interventions that inhibit TCTP’s functions are rational targets to be considered in cancer therapy. Regrettably, no attempts in this regard have been reported yet. Because of the diverse biological functions of TCTP such as development, growth, protein synthesis, and allergic response, targeting TCTP toward cancer cure should avoid systemic inhibition of this molecule and the potential risks to normal cells. A recent study of TCTP-induced negative regulation of autophagy has provided another mechanism for TCTP regulation of cancer progression. Since autophagy is the main machinery for the regulation of cellular homeostasis, TCTP also has the potential to affect cancer cell metabolism, which is different from conventional roles of TCTP in cancer, such as regulation of cancer cell proliferation. However, the consequences of targeting TCTP in cancer should be more precisely investigated because the role of autophagy in cancer is also dependent on the type and pathological stages of the specific tumor. In the early stages of tumorigenesis, which include initiation of malignant transformation and progression, autophagy may contribute to suppressing tumor development. However, in later stages, autophagy may help to sustain tumor growth by supporting energy production and by providing building blocks for cellular syntheses. Further mechanistic understanding of how TCTP mediates autophagy in cancer should help answer the question of whether TCTP can serve as a target in cancer therapy.

## Figures and Tables

**Figure 1 cells-09-00257-f001:**
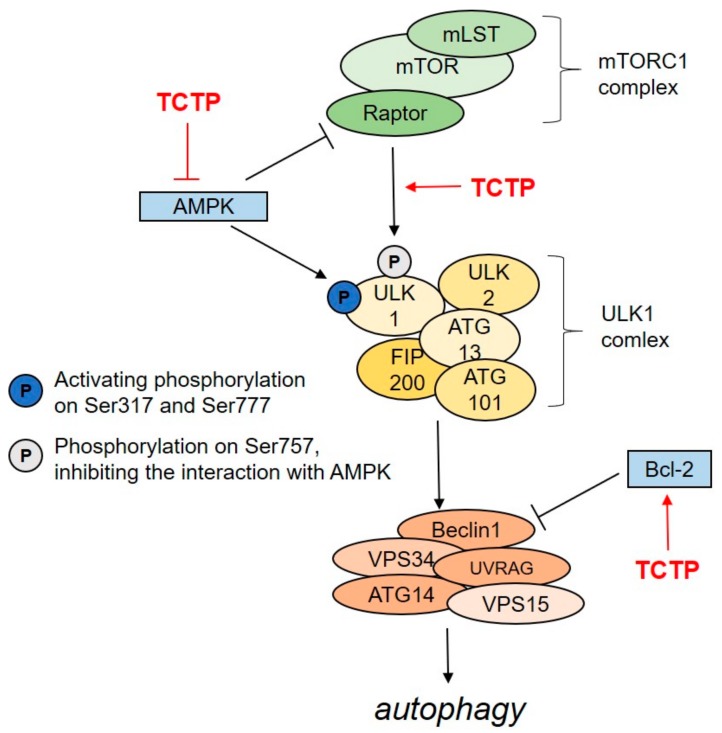
Negative regulation of autophagy by TCTP. In the initiation step of autophagy, TCTP may inhibit AMPK-induced ULK1 phosphorylation on Ser317 and Ser777, which activates the formation of the Beclin 1 complex. TCTP also enhances the mTORC1-induced phosphorylation of ULK1 on Ser757, which prevents ULK1 interaction with AMPK. TCTP also inhibits autophagosome formation by activating Bcl-2, which, in turn, inhibits the formation of the Beclin1 complex (see text for further details).

**Table 1 cells-09-00257-t001:** Roles of translationally controlled tumor protein (TCTP) in cancer pathogenesis.

Function	Mechanism	Reference
Anti-apoptosis	Increased Mcl-1 stability	[8]
	Inhibition of Bax dimerization	[9]
	Interaction with ATM and p53	[11,12]
	Promotion of MDM2-mediate p53 degradation	[22]
Pro-metastasis	Cdc42/JNK/MMP9 activation	[14]
	Regulation of HMGB1/NF-kB activation	[17]
	Promotion of EMT through mTORC2/Akt/GSK3β/β-catenin pathway	[15]
Resistance to anti-cancer therapy	Prevention of apoptosis through association with Apaf-1	[16]
	Translational upregulation of TCTP via mTORC1 pathway	[21]
	Promotion of MDM2-mediated p53 degradation	[24]

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
