# Peer review of "Regulation of Autophagy Is a Novel Tumorigenesis-Related Activity of Multifunctional Translationally Controlled Tumor Protein"

_cells, 2020, doi:10.3390/cells9010257_

Round 1

Reviewer 1 Report

In this review paper, authors focus on the role of translationally controlled tumor protein (TCTP) in autophagy. As author described, TCTP is known to be involved in anti-apoptotic function. However, their roles in autophagy lack evidence because of lower number of publication. I think that topic is interesting for the readers. Therefore, I suggest that authors should add the information of the role of TCTP in anti-apoptotic function.

Author Response

In this review paper, authors focus on the role of translationally controlled tumor protein (TCTP) in autophagy. As author described, TCTP is known to be involved in anti-apoptotic function. However, their roles in autophagy lack evidence because of lower number of publication. I think that topic is interesting for the readers. Therefore, I suggest that authors should add the information of the role of TCTP in anti-apoptotic function.

Response: The requested information about the role of TCTP in anti-apoptotic function has been provided in the manuscript (lines 26-38).

Reviewer 2 Report

Comments:

The occult relationship between cancer and autophagy has been gradually unraveled in the recent years. Thus targeting cancer by means of autophagy with different settings provides promising hopes for treatment. In this review, the authors focused on an interesting factor, TCTP, and discussed its role in cancer and newly discovered correlation with autophagy in multifaceted layers. While the review is overally comprehensive and well-organized, many details as listed below should be meliorated:

Line 86 “autophagy also has been considered as programmed cell death”.

Whether autophagy is considered as a form of cell death is largely controversial. See reference: Nat Rev Mol Cell Biol-2008-Kroemer-Autophagic cell death, the story of a misnomer.

In many cases, it is agreed that this autophagic cell death is cell death with autophagy rather than cell death by autophagy.

Line 115 “mouse model of ???”

What is the “???”

Line 119-121

Although the references [36, 37] match the content, the Keap1-mediated ubiquitination of Nrf2 was not mentioned in reference [38]. So [38] shouldn’t be cited here or to be replaced by another reference.

Line 121-124

The ref [39] is not so relevant to the claiming of “Another proposed mechanism is that autophagy is mediated by regulation of inflammation”. Should cite more relevant references introducing the relationship between autophagy and inflammation, such that Th1 cytokines, including IFN-y, TNF-a, IL-1, IL-6 have the effects of autophagy inducement, while the classical Th2 cytokines, including IL-4, IL-10 and IL-13, have the effects of inhibition, etc.

Line 124-126

There should have references following the sentence “Under chronic inflammation……and oncogenic transformation”.

Line 131

Wrongly wording of “(B)” at the beginning of this paragraph.

Line 153-154

There should have references following the sentence “During the initiation of cancer……genomic stability”.

Line 157

The word “both” should be included into the parenthesis.

Line 165

The reference [46] is too old. Should introduce more updated progress on the Tamoxifen-resistant breast cancer cells which have more actively induced autophagy. Accordingly, the concluding remark after [46] should be replaced.

Line 182

The word “persistent” is wrongly in the bold font.

Line 192-193

Incorrect conclusion. Bafilomycin A interferes the acidification of lysosome rather than the autophagosome-lysosome fusion as ref [56] mentioned; Spautin-1 inhibits autophagy by selectively promoting the degradation of the class III PI3 kinase complexes; Ref [58] is about Nedaplatin, which wasn’t even mentioned in this sentence after Bafilomycin A and Spautin-1.

Line 201

There should be a definition of “resistant mechanism” and “cell death mechanism” before using these highly summarized terms.

Line 204-252

There are indeed two different parts of content under this subtitle. The first paragraph from Line 205 to Line 226 is not about the role of TCTP in regulation of autophagy, is about how TCTP is possibly degraded by autophagy.

Thus, it’s better to divide this part into two subtitles: first paragraph goes to “5. Autophagy mediated degradation of TCTP”; second paragraph goes to “6. The role of TCTP in regulation of autophagy”.

Figure 2 on page 6

The caption for the grey “P” is incorrect. “Inhibiting phosphorylation on Ser757” should be changed to “Activating phosphorylation on Ser757”, which is the correct function of mTORC1 complex.

Author Response

The occult relationship between cancer and autophagy has been gradually unraveled in the recent years. Thus targeting cancer by means of autophagy with different settings provides promising hopes for treatment. In this review, the authors focused on an interesting factor, TCTP, and discussed its role in cancer and newly discovered correlation with autophagy in multifaceted layers. While the review is overally comprehensive and well-organized, many details as listed below should be meliorated:

Line 86 “autophagy also has been considered as programmed cell death”.

Response: Whether autophagy is to be considered as a form of cell death is controversial. See reference: Nat Rev Mol Cell Biol-2008-Kroemer-Autophagic cell death, the story of a misnomer.

In many cases, it is agreed that this autophagic cell death is cell death with autophagy rather than cell death by autophagy.

Response: We agree that we did not clearly state that autophagic cell death is cell death with autophagy rather than cell death by autophagy. We revised the revised manuscript as follows,

“Line 87: autophagy also has been classified as type II programmed cell death based on the observation of dying cells contain plenty of autophagic vacuolization in cytoplasm.”

“Line 89: However there is still debate on whether autophagic cell death is cell death accompanied by autophagy or cell death caused by autophagy [24].”

Line 115 “mouse model of ???”

What is the “???”

Response: We apologize for this typo. We corrected the manuscript.

“Line 118: Studies in mutant mouse model revealed beclin 1 as a haploinsufficient tumor suppressor gene [34,35] and provided evidence of negative regulation of tumorigenesis.”

Line 119-121

Although the references [36, 37] match the content, the Keap1-mediated ubiquitination of Nrf2 was not mentioned in reference [38]. So [38] shouldn’t be cited here or to be replaced by another reference.

 Response: We agree with the reviewer’s comment thus we removed citation [38].

Line 121-124

The ref [39] is not so relevant to the claiming of “Another proposed mechanism is that autophagy is mediated by regulation of inflammation”. Should cite more relevant references introducing the relationship between autophagy and inflammation, such that Th1 cytokines, including IFN-y, TNF-a, IL-1, IL-6 have the effects of autophagy inducement, while the classical Th2 cytokines, including IL-4, IL-10 and IL-13, have the effects of inhibition, etc.

Response: We intended to suggest inhibition of inflammation by autophagy as another mechanism of suppressing tumorigenesis and realized that original sentence might be misleading or confusing. So we revised the sentence in the manuscript as follows.

“Line 124: Another proposed mechanism is that autophagy suppresses tumorigenesis through inhibition of inflammation in cells.”

Ref [39] shows that when autophagy becomes deficient due to loss of Becn1 under the nutrition-limited condition, necrosis becomes associated with inflammation, as shown by increased macrophage infiltration and chemokine/cytokine production. Therefore this reference is relevant to the claim that autophagy can suppress tumorigenesis by inhibiting inflammation.

Line 124-126

There should have references following the sentence “Under chronic inflammation……and oncogenic transformation”.

Response: We added references about inflammation-induced tumorigenesis (line 129).

Line 131

Wrongly wording of “(B)” at the beginning of this paragraph.

Response: We typed (B) for additional explanation about autophagy’s role in the survival of cells which was mentioned in initial part of the former paragraph (line 114), but we now realize that these wordings are not necessary

Line 153-154

There should have references following the sentence “During the initiation of cancer……genomic stability”.

Response: We added references (line 159).

Line 157

The word “both” should be included into the parenthesis.

Response: We made the necessary correction in the revised manuscript (line 162).

Line 165

The reference [46] is too old. Should introduce more updated progress on the Tamoxifen-resistant breast cancer cells which have more actively induced autophagy. Accordingly, the concluding remark after [46] should be replaced.

Response: Instead of removing the reference 46, we added more recent studies about autophagy as a resistance mechanism in tamoxifen treatment (line 171).

Line 182

The word “persistent” is wrongly in the bold font.

Response: We changed the font (line 188).

Line 192-193

Incorrect conclusion. Bafilomycin A interferes the acidification of lysosome rather than the autophagosome-lysosome fusion as ref [56] mentioned; Spautin-1 inhibits autophagy by selectively promoting the degradation of the class III PI3 kinase complexes; Ref [58] is about Nedaplatin, which wasn’t even mentioned in this sentence after Bafilomycin A and Spautin-1.

Response: As reviewer mentioned, Bafilomycin A interrupts the acidification of lysosome. But this agent also blocks autophagy by blocking autophagosome-lysosome fusion (Caroline Mauvezin & Thomas P Neufeld (2015) Bafilomycin A1 disrupts autophagic flux by inhibiting both V-ATPase-dependent acidification and Ca-P60A/ SERCA-dependent autophagosome-lysosome fusion, Autophagy, 11:8, 1437-1438, DOI: 10.1080/15548627.2015.1066957 and Mauvezin, C. et al. Autophagosome–lysosome fusion is independent of V-ATPase-mediated acidification. Nat. Commun. 6:7007 doi: 10.1038/ncomms8007 (2015)). In the revised manuscript, we replaced the previous reference [56] to current reference [58]. Ref [59, previously 58] showed inhibition of autophagy by Bafilomycin A and 3-MA enhanced the antitumor efficacy of Nedaplatin, a cisplatin analog, thus this study can be a proper reference to cite potential of autophagy-inhibitory agent as an anticancer therapeutics. As reviewer suggested we corrected description of each agent as follows,

Line 198: Furthermore Bafilomycin A interferes with both lysosome acidification and autophagosome-lysosome fusion have the potential to be developed as anti-cancer agents to overcome the anticancer therapy resistance [62,63]. A potent autophagy inhibitor spautin-1 [64] also has been reported to enhance the potency of imatinib in chronic myeloid leukemia [65].”  

Line 201

There should be a definition of “resistant mechanism” and “cell death mechanism” before using these highly summarized terms.

Response: We amended original sentence as suggested.

Line 207:

Tyrosine kinase inhibitors and histone deacetylase inhibitors are also known to induce autophagy in cancer cells, but further investigations are needed on whether autophagy is induced by cells which are resistant to those inhibitors or as a readout of cell death.

Line 204-252

There are indeed two different parts of content under this subtitle. The first paragraph from Line 205 to Line 226 is not about the role of TCTP in regulation of autophagy, is about how TCTP is possibly degraded by autophagy.

Thus, it’s better to divide this part into two subtitles: first paragraph goes to “5. Autophagy mediated degradation of TCTP”; second paragraph goes to “6. The role of TCTP in regulation of autophagy”.

Response: we agree with the reviewer’s comment and divided this part as suggested (line 213 and 236).

Figure 2 on page 6

The caption for the grey “P” is incorrect. “Inhibiting phosphorylation on Ser757” should be changed to “Activating phosphorylation on Ser757”, which is the correct function of mTORC1 complex.

Response: The intended meaning of “inhibiting phosphorylation on Ser757” is that mTORC1 complex phosphorylates ULK1 at Ser757 which serves to inhibit autophagy by disrupting interaction between ULK1 and AMPK. But we realized that this expression might be misleading. We deleted “activiating” and inhibiting” to avoid such misleading.

Reviewer 3 Report

This is a very interesting review of the role of TCTP in autophagy.

Kyunglim Lee and colleagues provide with an exhaustive “tableau” on the subject, giving all details necessary. The paper is well written and of major interest to a wide audience.

I have not a single criticism on this article (what is unusual) and it can be published as is.

Author Response

N/A